# A genetic toolkit for tagging intronic MiMIC containing genes

Sonal Nagarkar-Jaiswal[1†], Steven Z DeLuca[2†], Pei-Tseng Lee[3], Wen-Wen Lin[3], Hongling Pan[1], Zhongyuan Zuo[3], Jiangxing Lv[3], Allan C Spradling[2], Hugo J Bellen[1,3,4,5,6*]

[1]Howard Hughes Medical Institute, Baylor College of Medicine, Houston, United States; [2]Department of Embryology, Howard Hughes Medical Institute, Carnegie Institution for Science, Baltimore, United States; [3]Department of Molecular and Human Genetics, Baylor College of Medicine, Houston, United States; [4]Jan and Dan Duncan Neurological Research Institute, Texas Children's Hospital, Houston, United States; [5]Department of Neuroscience, Baylor College of Medicine, Houston, United States; [6]Program in Developmental Biology, Baylor College of Medicine, Houston, United States

*For correspondence: hbellen@bcm.edu

†These authors contributed equally to this work

Competing interests: The authors declare that no competing interests exist.

**Abstract** Previously, we described a large collection of Minos-Mediated Integration Cassettes (MiMICs) that contain two phiC31 recombinase target sites and allow the generation of a new exon that encodes a protein tag when the MiMIC is inserted in a codon intron (*Nagarkar-Jaiswal et al., 2015*). These modified genes permit numerous applications including assessment of protein expression pattern, identification of protein interaction partners by immunoprecipitation followed by mass spec, and reversible removal of the tagged protein in any tissue. At present, these conversions remain time and labor-intensive as they require embryos to be injected with plasmid DNA containing the exon tag. In this study, we describe a simple and reliable genetic strategy to tag genes/proteins that contain MiMIC insertions using an integrated exon encoding GFP flanked by FRT sequences. We document the efficiency and tag 60 mostly uncharacterized genes.

## Introduction

One of the most powerful techniques for characterizing gene function is to generate transgenic animals in which an epitope tag such as GFP has been fused to the gene at its normal genomic location (*Ross-Macdonald et al., 1999*; *Morin et al., 2001*; *Skarnes et al., 2004*). These tagged proteins are extremely useful as they permit determination of protein localization in vivo as well as conditional, tissue specific, temporal and reversible removal of the tagged proteins (*Nagarkar-Jaiswal et al., 2015*). However, previous methods for generating protein trap alleles in *Drosophila* have allowed only about 800 genes to be successfully tagged (*Kelso et al., 2004*; *Buszczak et al., 2007*; *Quinones-Coello et al., 2007*; *Aleksic et al., 2009*; *Lowe et al., 2014*).

We previously developed a flexible system for engineering the *Drosophila* genome using the Minos-Mediated Integration Cassette (MiMIC) transposable element. We generated 15,660 strains with a single MiMIC inserted at random within the fly genome and mapped their insertion site (*Bellen et al., 2011*; *Venken et al., 2011*; *Nagarkar-Jaiswal et al., 2015*). MiMIC carries sequences that function as a gene and protein trap when inserted in the proper orientation in a coding intron. Moreover, its content can be replaced by Recombination-Mediated Cassette Exchange (RMCE) leading to the introduction of any desired DNA, such as an artificial exon that encodes a protein tag. This approach can potentially be used to tag thousands of genes. Currently, 2854 existing insertions

are located within the coding introns of 1862 distinct genes (*Nagarkar-Jaiswal et al., 2015*), and MiMIC-like elements can now be placed in any gene of interest by CRISPR (*Zhang et al., 2014*). Unfortunately, the RMCE method needed to convert these insertions into functional protein traps requires embryonic injections of an appropriate donor DNA and screening of many offspring to identify the desired events, a labor and cost-intensive procedure that does not scale easily. We therefore developed a more efficient and economical in vivo genetic tagging methodology that can in principle be used to generate protein trap alleles of all *Drosophila* genes.

## Results and discussion

We developed a genetic strategy that allows the desired RMCE event to take place efficiently without the need for microinjection. The method uses FLP recombinase to release a genomically integrated DNA flanked by FRT sites into the nucleoplasm where it can efficiently undergo phiC31 integrase-mediated cassette exchange, as shown by *Gohl et al. (2011)*. As shown in *Figure 1A*, we engineered three donor cassettes, one for each reading frame. The core, which contains a splice acceptor (SA) followed by a (GGS)$_4$ flexible linker, multiple tags (EGFP-FlAsH-StrepII-TEVcs-3xFlag {GFSTF}), another (GGS)$_4$ flexible linker, and a splice donor (SD), is flanked by two inverted *attB* sites for phiC31-mediated RMCE (*Venken et al., 2011*). We then cloned this cassette core between tandem FRT sites in a *P-element* transformation vector (*Gong and Golic, 2003*). FLP-mediated recombination between the tandem FRT sites excises a circular donor DNA molecule from its initial genetic locus, promoting its efficient recombination with a distal target site (*Golic et al., 1997*). A mini-white eye color marker gene between our donor cassette and one of the FRT sites allows us to monitor the presence or absence of the donor cassette in FLP recombinase-containing stocks.

We created 6 stocks (*Figure 1—source data 2*), each harboring one of the three donor transgenes located on the second or third chromosome, and a heat shock-inducible FLP recombinase and a germ line-expressed phiC31 integrase on the X-chromosome. Because the heat shock-inducible FLP recombinase is somewhat leaky at 18°C, the donor transgene is lost from these stocks at a low frequency, resulting in rare white-eyed flies, which we periodically discard.

To initiate RMCE, we crossed the appropriate donor flies to MiMIC-containing flies and heat shocked the resulting embryos and larvae (*Figure 2*). Within the primordial germ cells of some of the embryos and larvae, phiC31 integrase catalyzed recombination between *attB* sites in the donor and *attP* sites in the MiMIC transposon. The positive RMCE events were selected based on the loss of the y + marker present in the original MiMIC (*Figure 2*). We confirmed the integration and orientation of the donor cassette by PCR as described in *Venken et al. (2011)*. Typically, 50% of the integration events are in the proper orientation.

We observed one to ten RMCE events in 93 out of 113 attempts in our initial trial when we set up 3–7 crosses (Cross 2 in *Figure 2*). After PCR screening, 60/93 of the tested MiMICs allowed integration of at least one donor in the proper orientation to tag the endogenous gene (*Supplementary file 1*). In summary, we set up 3–7 vials for each starting cross and obtained 60/113 tagged genes. Since the efficiency of RMCE and the ease of detecting yellow⁻ progeny vary between different starting sites, we propose to set up 10–20 vials and to score more progeny to improve the success rate. The method has been found to work for a wide variety of genes including a gene located in a telomeric region (*lethal giant larvae (l(2)gl)*), suggesting that there may be few limitations in its applicability.

To ensure that the expression pattern and protein distribution correspond to the endogenous protein, we costained two tagged lines with GFP for which specific monoclonal antibodies are available: Eyes shut (Eys) (mAb 21A6,) and Delta (Dl) (mAb C594.9B) (*Figure 3A*). In both cases, the protein recognized by the mAb colocalizes with the GFP and match the described expression patterns (*Das et al., 2013*; *Haltom et al., 2014*). Note, however, that the GFP tagged Eys protein is present in the cytoplasm of the photoreceptors and the inter-rhabdomere spaces (IRS) of the photoreceptors, whereas the mAb against Eys mostly localizes to the IRS (*Figure 3A*). These data are in agreement with what we previously observed for numerous tagged proteins (*Venken et al., 2011*; *Nagarkar-Jaiswal et al., 2015*).

We stained third instar larval brains and discs for the 60 tagged gene/proteins. The examples, shown in *Figure 3B*, include *lethal (2) giant larvae (l(2)gl)* (a), *Delta (Dl)* (b), and *twins (tws)* (c) whose expression patterns are consistent with published data (*Kooh et al., 1993*; *Albertson and Doe, 2003*; *Chabu and Doe, 2009*). Similarly, *kayak/fos (kay)* is expressed in wing disc nuclei (d) as described earlier (*Zeitlinger and Bohmann, 1999*). The expression pattern of the remaining genes has not been previously described (*Figure 3B*): *Saposin-related (Sap-r)* is expressed in a subset of cells in larval

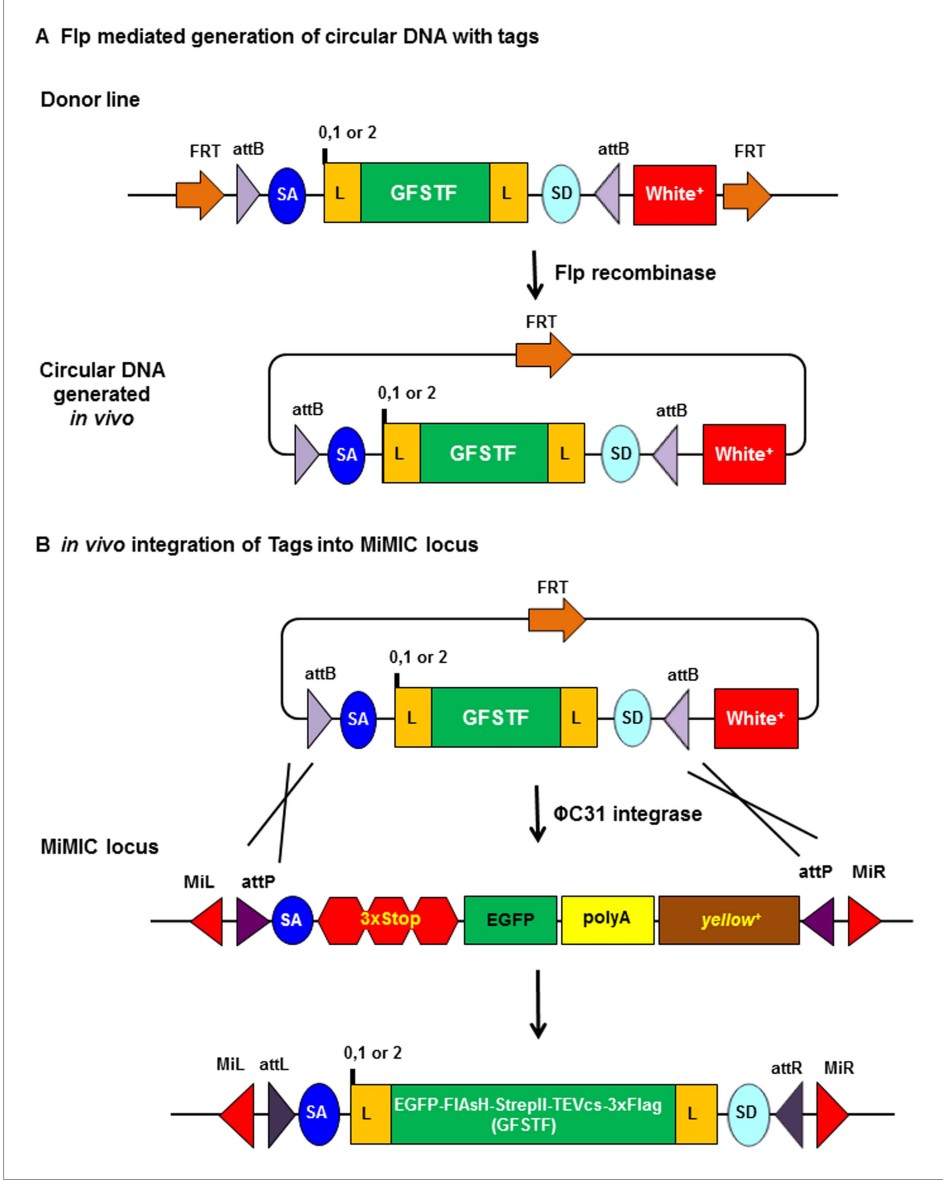

**Figure 1**. Schematic of flippase and phiC31 integrase mediated in vivo protein tagging. (**A**) Flp mediated generation of circular DNA with tag sequences. Donor line carries a source of Flp recombinase and phiC31 integrase, and the core cassette for a specific phase (0, 1, or 2) encoding a $(GGS)_4$ flexible linker (L), multiple tags EGFP-FlAsH-StrepII-TEVcs-3xFlag (GFSTF), another $(GGS)_4$ flexible linker, and a splice donor (SD), which is flanked by two inverted *attB* and two FRT sites oriented in that same direction. Upon Flp expression, the core is flipped out as a circular DNA containing one FRT site. (**B**) In vivo integration of tag sequence into Minos-Mediated Integration Cassette (MiMIC) locus. At a MiMIC locus in a coding intron, the MiMIC gene trap cassette is replaced by L-EGFP-FlAsH-StrepII-TEVcs-3xFlag-L sequence by phiC31 integrase resulting in loss of *yellow+* marker.

The following source data are available for figure 1:

**Source data 1**. List of constructs.
**Source data 2**. List of fly strains generated.

brain (e), *Rad, Gem/Kir family member 3* (*Rgk3*) is enriched in mushroom body in L3 larval brain (f), *Heterogeneous nuclear ribonucleoprotein at 98DE* (*Hrb98DE*) is expressed in L3 larval brain (g), *CG10086* is expressed in hindgut (h), and *CG5656* is expressed in the cells of the cuticle (i).

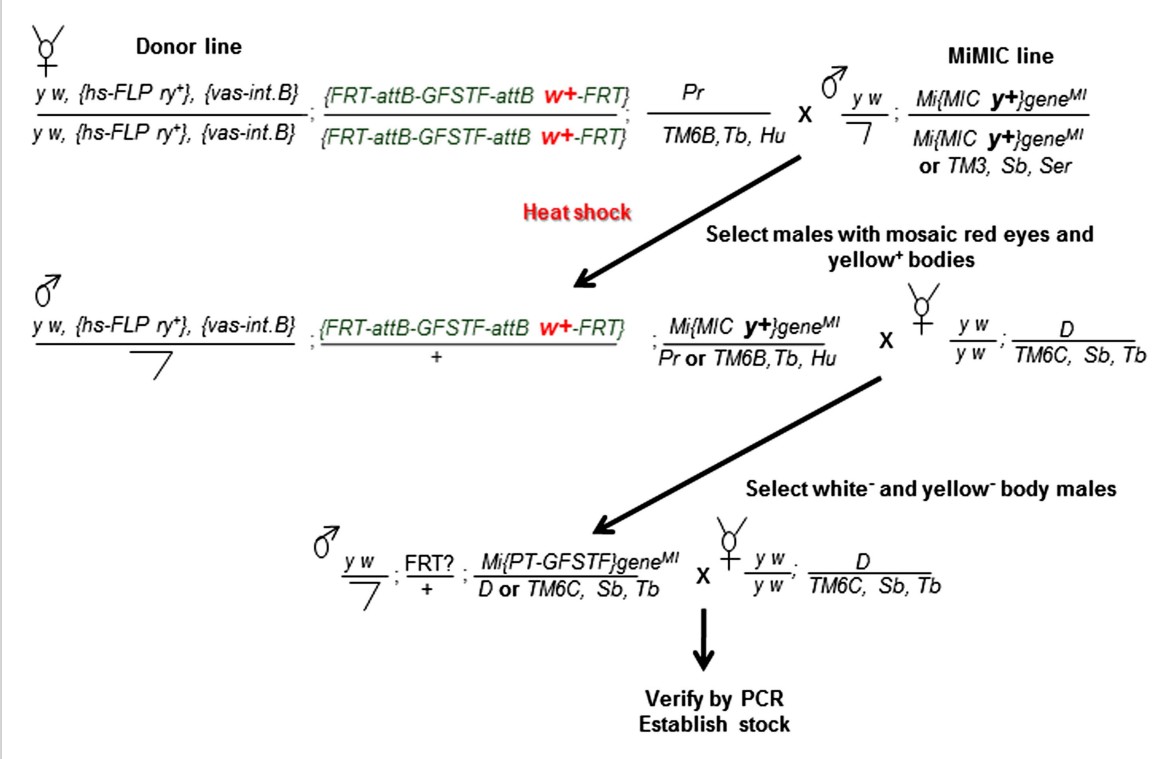

**Figure 2**. Crossing scheme for generating EGFP tagged MiMIC lines. Females carrying the hs-FLP and *vasa-phiC31* integrase on the X-chromosome, and the FRT flanked multiple tag (GFSTF) cassette on the II-chromosome are crossed to males carrying a MiMIC insertion in a coding intron of a gene on the III-chromosome. The resulting embryos are heat shocked, and adult progeny with the desired genotype are crossed with flies carrying appropriate balancers. The resulting progeny is screened for the loss of *white+* and *yellow+* and crossed to flies carrying appropriate balancers to establish stocks, which are then verified by PCR assay.

The expression patterns of all these genes as well as all the genes listed in *Supplementary file 1* are documented in the MiMIC RMCE database at http://flypush.imgen.bcm.tmc.edu/pscreen/rmce/.

In summary, we developed a genetic tagging strategy that will greatly facilitate the EGFP tagging of nearly 2000 genes that already carry MiMIC insertions (*Nagarkar-Jaiswal et al., 2015*). The same strategies can also be used for tagging genes with other protein tags. In addition, a similar strategy based on *lox* sites instead of *FRT* cassettes has recently been developed to integrate an artificial exon carrying the *GAL4* gene in MiMICs inserted in coding introns (*Diao et al., 2015*). These insertions are mutagenic but permit the expression of the endogenous wild-type and mutant cDNAs of *Drosophila* as well as other species under the control of UAS. Moreover, these tagging methods can now be combined with CRISPR directed integration of *attP* carrying cassettes similar to MiMIC in coding introns to tag almost every gene in *Drosophila* (*Zhang et al., 2014*).

## Material and methods

### Cloning

The core cassettes (attB-SA- phase 0/1/2-[GGS]$_4$-EGFP-FlAsH-StrepII-TEVcs-3xFlag-[GGS]$_4$-SD-attB) for Phase 0,1, or 2 were excised from pBS-KS-attB1-2-PT-SA-SD-0-EGFP-FlAsH-StrepII-TEVcs-3xFlag, pBS-KS-attB1-2-PT-SA-SD-1-EGFP-FlAsH-StrepII-TEVcs-3xFlag, or pBS-KS-attB1-2-PT-SA-SD-2-EGFP-FlAsH-StrepII-TEVcs-3xFlag as NheI/NsiI fragments and subcloned into *P-element* vector pW35 (DGRC) between PstI/AvrII to create final donor vectors pW35-FRT-attB-SA-phase 0-(GGS)$_4$-EGFP-FlAsH-StrepII-TEVcs-3xFlag-(GGS)$_4$-SD-attB-white+-FRT, pW35-FRT-attB-SA-phase 1-(GGS)$_4$-EGFP-FlAsH-StrepII-TEVcs-3xFlag-(GGS)$_4$-SD-attB-white+-FRT and pW35-FRT-attB-SA-phase 2-(GGS)$_4$-EGFP-FlAsH-StrepII-TEVcs-3xFlag-(GGS)$_4$-SD-attB-white+-FRT.

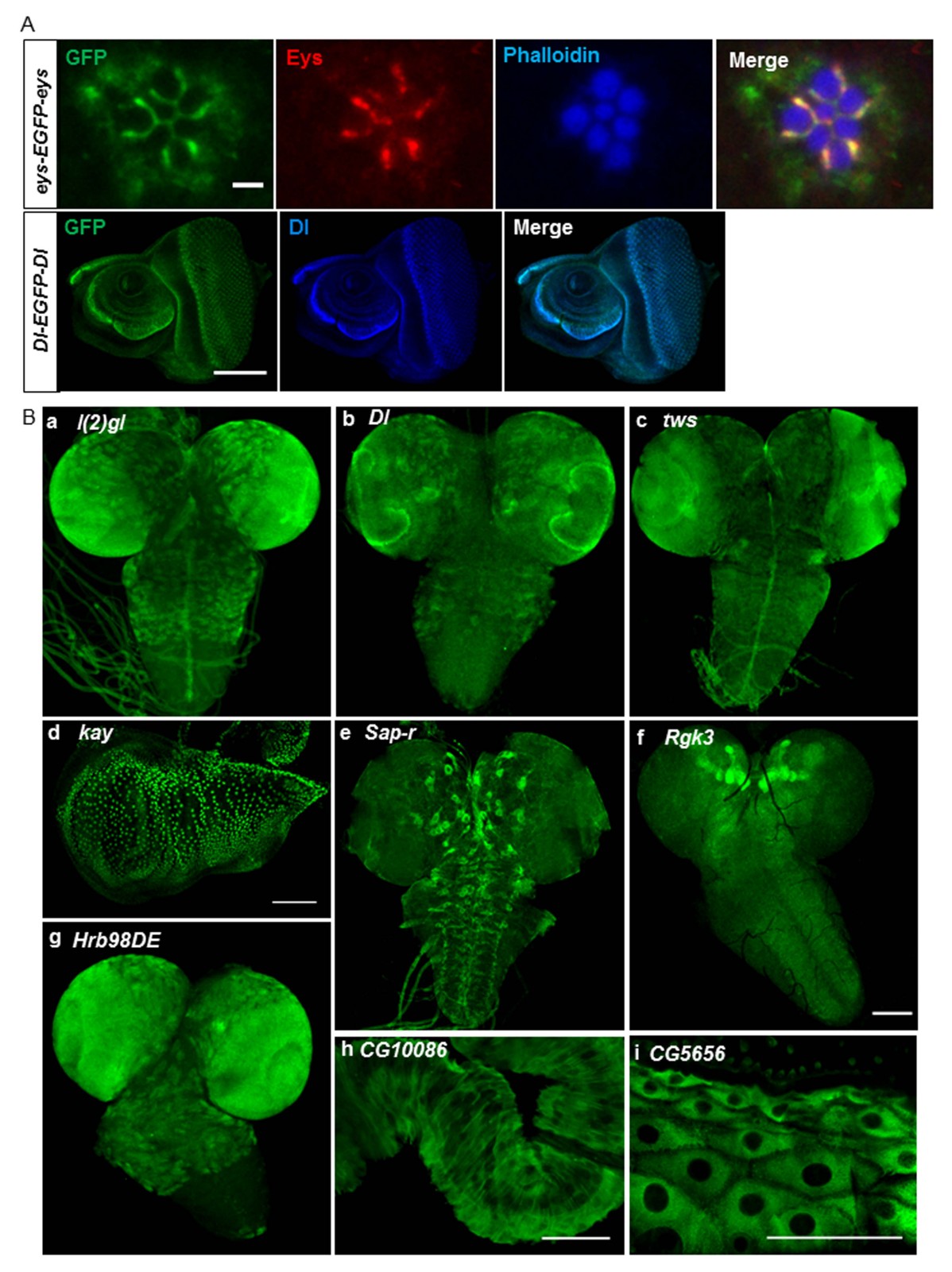

**Figure 3**. Expression of EGFP tagged protein in various tissues. (**A**) Colocalization of GFP tagged proteins with gene specific antibodies: anti-GFP signals (green) show colocalization with anti-Eys (red) in inter-rhabdomere space in adult eye (top panel) and anti-Delta (blue) in L3 eye imaginal disc in bottom panel. Scale bars, 2 μm and 5 μm. (**B**) Examples of EGFP expression pattern in different tissues from third instar larvae: brain; l(2)gl (**A**), Dl (**B**), twins (tws) (**C**), Sap-r (**E**), Rgk3 (**F**), and Hrb98DE (**G**), wing imaginal disc: kay (**D**), hindgut: CG10086 (**H**), and cells of the cuticle: CG5656 (**I**). Scale bars 100 μm. (**A–I**, except **G**), 25 μm (**G**). Eys, eyes shut.

## In vivo tagging

Around 15–20 phase-specific homozygous donor females ($P\{ry^{+t7.2}= hsFLP\}12$, $y^1w^*M\{vas\text{-}int.B\}ZH\text{-}2A$; $P\{FRT\text{-}attB\text{-}\{GFSTF\}\text{-}attB(w^+)\text{-}FRT\}$; $Pri^1/TM6B$, $Tb^1$ or $P\{ry^{+t7.2} = hsFLP\}12$, $y^1$ $w^*M\{vas\text{-}int.B\}ZH\text{-}2A$; $S^1/CyO$; $P\{FRT\text{-}attB\text{-}\{GFSTF\}\text{-}attB (w^+)\text{-}FRT\}$)) were crossed with 5–10 males carrying MiMIC insertion in coding intron ($y^1 w^*$; $Mi[MIC$ $y^+]$ $gene^{MI}/Mi[MIC$ $y^+]$ $gene^{MI}$ or balancer). Crosses were transferred to new vials every third day and constantly kept at 18°C. Vials with progeny embryos and larvae were heat shocked on day 3, 4, and 5 for 20 minutes at 37°C in a water bath and raised at 25°C. About 5–7 vials with a pool of 5 F1 males with mosaic red eyes and yellow body were crossed with 10–15 virgins of $y^1w^{67c23}$; $In(2LR)Gla$, $wg^{Gla-1}/SM6a$ or $y^1w^*$; $D/TM6b$, $Hu$, $Tb$. Transgenic F2 progeny were screened for loss of $yellow^+$ (yellow-phenotype) and subsequently crossed to virgins of $y^1w^{67c23}$; $In(2LR)Gla$, $wg^{Gla-1}/SM6a$ or $y^1w^*$; $D/TM3,Sb$, $Tb$ to establish stocks. Correct RMCE events were confirmed by PCR assay as described in *Nagarkar-Jaiswal et al., 2015*.

## Immunostaining

Briefly, third instar larvae were dissected for larval brains, imaginal discs, and gut in 1xPBS and fixed in 3.7% formaldehyde for 30 min at room temperature and washed in 0.2% Triton X-100 (*Nagarkar-Jaiswal et al., 2015*). They were then incubated for 1 hr at RT in 10% NGS-PBS-0.2% Triton X-100 and stained with primary antibodies diluted in 10% NGS-PBS-0.2% Triton X-100 overnight at 4°C. The samples were washed and incubated with secondary antibodies for 2 hr at RT. The samples were then washed and mounted in Vectashield (Vector Labs, Burlingame, CA) and imaged with a Zeiss LSM710 confocal microscope and processed using Adobe Photoshop (Adobe Systems Inc., San Jose, CA, USA).

## Antibodies used

Primary antibodies used: rabbit anti-GFP 1:1000 (Life Technologies, A11122), mouse anti-Delta 1:1000 (C594.9B, DSHB [*Qi et al., 1999*]), and mouse anti-Eys 1:250 (21A6, DSHB [*Fujita et al., 1982*]). Secondary antibodies used: Alexa 488 (Invitrogen, Life Technologies, Grand Island, NY), Cy5 and Cy3 conjugated antibodies (Jackson ImmunoResearch, West Grove, PA) were used at 1:500.

# Acknowledgements

We thank Qiaohong Gao, Zhihua Wang, and Paolo Mangahas for technical help. We thank Karen L Schulze and Megan E Campbell for comments on the manuscript. This research was supported by NIGMS R01GM067858. Confocal microscopy was supported by NICHD 1U54HD083092 to the Baylor College of Medicine Intellectual and Developmental Disabilities Research Center. We thank the Bloomington *Drosophila* Stock Center (BDSC) for numerous stocks. HJB and AS are Investigators of the Howard Hughes Medical Institute. SZD is supported by a Helen Hay Whitney Foundation postdoctoral fellowship.

# Additional information

### Funding

| Funder | Grant reference | Author |
| --- | --- | --- |
| National Institute of General Medical Sciences (NIGMS) | RO1GM067858 | Pei-Tseng Lee, Wen-Wen Lin, Zhongyuan Zuo, Jiangxing Lv |
| Howard Hughes Medical Institute (HHMI) | HHMI | Sonal Nagarkar-Jaiswal, Hongling Pan, Allan C Spradling, Hugo J Bellen |
| Helen Hay Whitney Foundation | Postdoctoral fellowship | Steven Z DeLuca |

The funders had no role in study design, data collection and interpretation, or the decision to submit the work for publication.

## Author contributions

SN-J, Conception and design, Acquisition of data, Analysis and interpretation of data, Drafting or revising the article; SZDL, ACS, HJB, Conception and design, Drafting or revising the article; P-TL, W-WL, HP, ZZ, JL, Acquisition of data, Drafting or revising the article

## Additional files

### Supplementary file

• Supplementary file 1. Total EGFP tagged genes created: MI: MiMIC insertion, GT: Gene Trap, PT: Protein Trap, Y/N: Yes/No, and L/V: Lethal/Viable

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
