## [Decision Letter]

Thank you for submitting your work entitled “A genetic toolkit for tagging intronic MiMIC containing genes” for peer review at *eLife*. Your submission has been favorably evaluated by K VijayRaghavan (Senior editor), Mani Ramaswami (Reviewing editor), and two reviewers.

The reviewers have discussed the reviews with one another and the Reviewing editor has drafted this decision to help you prepare a revised submission.

Summary:

This Research Advance describes the creation of a genetic toolkit that allows for the (Recombination Mediated Cassette Exchange) RMCE of an integrated exon encoding for GFP with existing MIMIC insertions. This resource bypasses the need for laborious injections and will facilitate the use of MIMIC lines for epitope tagging genes at their endogenous sites. An extension of the recent MiMIC resource that was published in *eLife*, it documents a simplified and streamlined version of converting an intronic MiMIC insertion (containing 2 *attP* sites) into a GFP tagged protein locus without need for the traditional labour-intensive injection approach. The source of the circular donor plasmid that contains both *attB* sites and the GFP cassette is a Flp mediated excision of a transgenic insertion. Transgenic lines for all three reading frames on second and third chromosome together with Flp and phiC31 sources were generated. The successful exchange has been demonstrated for 60 loci.

This RMCE approach provides a significant advance of the former MiMIC resource and can also be used beyond MiMIC, e.g. if the *attP* cassette has been inserted by CRISPR.

Essential revisions:

Several points should be addressed:

1) Figure 1 appears incorrect: surely the *white*^*+*^ box should be between the *attB* site and the FRT.

2) In the paragraph starting with “We observed RMCE events…”, the sentence “We set up 3-7 vials for each starting cross and this was sufficient to obtain 60/116 (113 or 116) tagged genes.” What is meant by (113 or 116)? This paragraph suggests that RMCE is unsuccessful in close to half of the lines tested. Do the authors have any insights into why this may be the case? Is there a common feature amongst the MIMIC lines that failed RMCE?

The authors should provide some idea of how efficient the process is. How many individual events were obtained from the 3-7 starting vials? How many flies were typically screened to get potential recombinants?

3) In “Since the efficiency of RMCE…”, yellow should be replaced by *yellow*^*–*^.

4) In the same paragraph, the authors should list examples of genes in heterochromatic regions.

5) Given the utility of newly developed CRISPR/Cas9 techniques, the authors should discuss the strengths of MIMIC system. The case can certainly be made. But the GFP cassettes described here represent only an incremental start. This resource would be greatly enhanced by generating lines with additional tags, as previously described by the authors.

---

## [Author Response]

*1)*
Figure 1
*appears incorrect: surely the* white^+^
*box should be between the* attB *site and the FRT*.

We thank the reviewers for pointing out this error. We made the correction in Figure 1.

*2) In the paragraph starting with* “*We observed RMCE events…*”*, the sentence* “*We set up 3-7 vials for each starting cross and this was sufficient to obtain 60/116 (113 or 116) tagged genes.” What is meant by (113 or 116)? This paragraph suggests that RMCE is unsuccessful in close to half of the lines tested. Do the authors have any insights into why this may be the case? Is there a common feature amongst the MIMIC lines that failed RMCE?*

We thank the reviewers for pointing out this issue. We have deleted the numbers.

*This paragraph suggests that RMCE is unsuccessful in close to half of the lines tested. Do the authors have any insights into why this may be the case? Is there a common feature amongst the MIMIC lines that failed RMCE? The authors should provide some idea of how efficient the process is*.

The RMCE was successful in 82% (93/113) of cases (please see the first sentence of this paragraph). This number includes EGFP cassette insertions in positive and negative orientation. However the probability of getting EGFP inserted in the correct orientation is statistically 50% of total RMCE events. The number 60 is the number of genes for which the EGFP cassette was inserted in the correct orientation resulting in EGFP tagged fusion proteins. Therefore with the current number of crosses (3-7 vials; Cross 2 in Figure 2) the efficiency is 53% (60/113). This can be increased by setting a greater number of screening crosses (10-20 vials per line) to obtain more RMCE events with insertion in positive orientation. Indeed, we recovered significantly more positive events with 7 crosses than with three crosses.

How many individual events were obtained from the 3-7 starting vials? How many flies were typically screened to get potential recombinants?

We obtained 1-10 RMCE positive individuals after screening a total of 300-700 flies from 3-7 vials.

*3) In* “*Since the efficiency of RMCE…*”*, yellow should be replaced by* yellow^–^.

We have corrected it.

*4) In the same paragraph, the authors should list examples of genes in heterochromatic regions*.

We have included the gene name (*Lethal giant larva (l(2)gl)*) in text.

*5) Given the utility of newly developed CRISPR/Cas9 techniques, the authors should discuss the strengths of MIMIC system. The case can certainly be made. But the GFP cassettes described here represent only an incremental start. This resource would be greatly enhanced by generating lines with additional tags, as previously described by the authors*.

We have modified the last paragraph of the Discussion.